# STOCHASTIC ADVERSARIAL NETWORK FOR MULTI-DOMAIN TEXT CLASSIFICATION

## ABSTRACT

Adversarial training has played a pivotal role in the significant advancements of multi-domain text classification (MDTC). Recent MDTC methods often adopt the shared-private paradigm, wherein a shared feature extractor captures domain-invariant knowledge, while private feature extractors per domain extract domain-dependent knowledge. These approaches have demonstrated state-of-the-art performance. However, a major challenge remains: the exponential increase in model parameters as new domains emerge. To address this challenge, we propose the Stochastic Adversarial Network (SAN), which models multiple domain-specific feature extractors as a multivariate Gaussian distribution rather than weight vectors. With SAN, we can sample as many domain-specific feature extractors as necessary without drastically increasing the number of model parameters. Consequently, the model size of SAN remains comparable to having a single domain-specific feature extractor when data from multiple domains. Additionally, we incorporate domain label smoothing and robust pseudo-label regularization techniques to enhance the stability of the adversarial training and improve feature discriminability, respectively. The evaluations conducted on two prominent MDTC benchmarks validate the competitiveness of our proposed SAN method against state-of-the-art approaches.

## 1 INTRODUCTION

Text classification has garnered considerable attention within Natural Language Processing (NLP) (Khurana et al., 2023). Over the past decade, deep learning has propelled text classification forward, albeit at the expense of requiring extensive labeled data (Kowsari et al., 2019). However, it is widely acknowledged that text classification is highly dependent on the specific domain. In other words, the same word can convey different sentiments across different domains (Wu et al., 2022b). This can easily result in a model trained in one domain easily performing poorly when applied to another domain. Unfortunately, collecting a substantial amount of labeled data for each desired domain is often prohibitively expensive and unrealistic. Thus, it becomes crucial to investigate approaches for leveraging knowledge from related domains to enhance the classification accuracy in the target domain.

Multi-domain text classification (MDTC) is proposed to address the problem stated above (Li & Zong, 2008). Earlier MDTC methods employed a per-domain training approach and utilized ensemble learning strategies to generate final results (Li et al., 2012; Wu & Huang, 2015). The most recent MDTC approaches can yield state-of-the-art performance by adopting adversarial training (Creswell et al., 2018; Ganin et al., 2016) and the shared-private scheme (Bousmalis et al., 2016b). Adversarial training aligns different domains to extract domain-invariant features, while the shared-private scheme partitions the latent space into a shared component that captures common features across domains, and multiple domain-specific feature spaces that capture domain-unique features. The domain-invariant features are expected to be both discriminative and transferable, whereas the domain-specific features enhance the discriminability of the domain-invariant features (Bousmalis et al., 2016a). However, these approaches face a challenge: the shared-private paradigm requires training domain-specific feature extractors for each domain, which often involves complex neural network architectures. As new domains emerge, incorporating numerous domain-specific feature extractors not only increases the number of model parameters (as depicted in Table 1), but also hampers training convergence.

Table 1: The number of parameters of the shared feature extractor $F_s$, the domain-specific feature extractors $\{F_d^i\}_{i=1}^M$, the classifier $C$, and the domain discriminator $D$ in MAN (Chen & Cardie, 2018) on different tasks. Obviously, the domain-specific feature extractor parameters take the majority part in both tasks, demonstrating that tackling data from more domains in MDTC will drastically increase the model size.

| Task | Amazon | FDU-MTL |
|---|---|---|
| # Para. of $F_s$ | 5.57M | 20.20M |
| # Para. of $\{F_d^i\}_{i=1}^M$ | 22.13M=4*5.57M | 322.65M=16*20.20M |
| # Para. of $C$ | 0.04M | 0.04M |
| # Para. of $D$ | 0.02M | 0.02M |
| # Total Para. | 27.76M | 342.91M |

To mitigate the aforementioned issue, we propose a novel approach called Stochastic Adversarial Network (SAN) that introduces a stochastic feature extractor to replace multiple domain-specific feature extractors. The stochastic feature extractor seamlessly integrates an infinite number of domain-specific feature extractors into existing MDTC methods, while keeping the model parameters unchanged. In SAN, instead of specific weight points used in previous MDTC approaches, the domain-specific feature extractors are represented by a weight distribution. Specifically, we model the domain-specific feature extractors using a Gaussian distribution, with the mean representing the final domain-specific feature extractor weight and the variance capturing the discrepancy among different domains. During training, the domain-specific feature extractor is sampled from the current distribution estimation, and the Gaussian distribution is optimized throughout the training. Consequently, the SAN model can extract domain-specific features across multiple domains using only one domain-specific feature extractor. Notably, this is achieved without the need to consider the number of required domain-specific feature extractors, while avoiding the negative impact of increasing the model size. To further enhance model performance, we incorporate domain label smoothing and robust pseudo-label regularization into the SAN method, ensuring stability in the adversarial training and improving feature discriminability. Through experiments conducted on two MDTC benchmarks, we demonstrate the effectiveness of our SAN approach, achieving competitive performance compared to state-of-the-art methods.

Our contributions are summarized as follows:

- We propose the Stochastic Adversarial Network (SAN) for MDTC, introducing a stochastic feature extractor mechanism. This enables MDTC models to learn domain-specific features from multiple domains using a single domain-specific feature extractor, thereby substantially reducing the number of model parameters. To the best of our knowledge, this study represents the first exploration of this matter in MDTC.

- We incorporate domain label smoothing and robust pseudo-label regularization techniques to stabilize the adversarial training and enhance the discriminability of the acquired features, respectively.

- The experimental results on two benchmarks illustrate the efficacy of the SAN method in comparison to state-of-the-art approaches. Additionally, we perform extensive experiments on multi-source unsupervised domain adaptation to highlight the generalization ability of our proposed SAN approach.

## 2 RELATED WORK

**Adversarial Training (AT).** AT, initially introduced by the Generative Adversarial Network (GAN) (Creswell et al., 2018) for image generation, involves a generator synthesizing images and a discriminator distinguishing between generated and real images. Domain-Adversarial Neural Networks (DANN) (Ganin et al., 2016) apply AT to domain adaptation by training a feature extractor against a domain discriminator. The domain discriminator aims to distinguish source and target features, while the feature extractor aims to deceive the domain discriminator, generating domain-invariant features when the discriminator cannot discern the feature source. Conditional Adversarial Neural Networks

(CDANs) (Long et al., 2018) employ multilinear conditioning to align conditional distributions and incorporate entropy conditioning to facilitate transfer learning. However, AT often exhibits oscillatory gradients during training, resulting in instability, slow convergence, and mode collapse (Arjovsky & Bottou, 2017; Mescheder et al., 2018). To overcome these limitations, Wasserstein GAN (Arjovsky et al., 2017) employs the earth mover distance to measure domain divergence. Additionally, Environment Label Smoothing (ELS) (Zhang et al., 2023) encourages the domain discriminator to output soft probabilities, enhancing the stability of AT.

**Stochastic Neural Network (SNN).** The weight parameters of a neural network are typically treated as point estimates, limiting their ability to capture uncertainty and often resulting in overconfident predictions (Blundell et al., 2015). To address this limitation, SNNs are proposed, which consider weight parameters as random variables sampled from specific distributions. For example, Bayesian Neural Networks (BNNs) (Hernández-Lobato & Adams, 2015; Wang & Yeung, 2020) are widely used to represent intermediate outputs and final predictions as stochastic variables, providing richer representations. The Auto-Encoding Variational Bayes (AEVB) (Kingma & Welling, 2013) employs a Gaussian distribution to model latent variables in image inputs, serving as a form of data augmentation. Uncertainty-aware multi-modal BNNs (Subedar et al., 2019) combine deterministic and variational layers for activity recognition, while DistributionNet (Yu et al., 2019) models feature uncertainty in person re-identification using distributions. In unsupervised domain adaptation, the Stochastic Classifier (Lu et al., 2020) leverages a Gaussian distribution to model classifier parameters.

**Multi-domain text classifications (MDTC).** MDTC aims to enhance overall classification accuracy by harnessing available resources from multiple domains (Li & Zong, 2008). Early MDTC methods employ transfer learning techniques to drive progress. The structural correspondence learning (SCL) (Blitzer et al., 2006) method computes relationships between different pivot features to learn correspondences among them. The collaborative multi-domain sentiment classification (CMSC) (Wu & Huang, 2015) method trains two types of classifiers: a shared classifier for all domains and a set of domain-specific classifiers for each domain, combining their outputs for final results. Recent MDTC approaches commonly adopt the adversarial training and shared-private paradigm, leading to significant advancements. The domain separation network (DSN) (Bousmalis et al., 2016a) first introduces the shared-private paradigm for adversarial domain adaptation and empirically demonstrates that domain-unique features can enhance the discriminability of domain-invariant features. The adversarial multi-task learning (ASP-MTL) method (Liu et al., 2017) applies adversarial training and the shared-private paradigm to MDTC. The multinomial adversarial networks (MANs) (Chen & Cardie, 2018) utilize the least square loss and negative log-likelihood loss to train the domain discriminator. The mixup regularized adversarial networks (MRANs) (Wu et al., 2021b) propose domain and category mixup regularizers for MDTC. The maximum batch Frobenius norm (MBF) (Wu et al., 2022b) method improves feature discriminability by maximizing the Frobenius norm of the intermediate feature matrix.

In contrast to previous MDTC approaches that utilize separate domain-specific feature extractors for each domain, our proposed SAN method employs parameter sampling from a Gaussian distribution to model the domain-specific feature extractor. This approach enables the SAN method to acquire domain-specific knowledge through a single feature extractor, resulting in a significant reduction in the number of model parameters needed.

## 3 METHOD

The MDTC task can be formulated as follows: given $M$ domains $\{D_i\}_{i=1}^M$, each domain contains a small amount of labeled data $\mathbb{L}_i = \{\mathbf{x}_j, y_j\}_{j=1}^{l_i}$ and a large amount of unlabeled data $\mathbb{U}_i = \{\mathbf{x}_j\}_{j=1}^{u_i}$. The primary objective of MDTC is to leverage these resources to enhance the average classification accuracy across all domains.

### 3.1 ADVERSARIAL MULTI-DOMAIN TEXT CLASSIFICATION

Adversarial training has proven to be effective in mitigating domain discrepancies and has found widespread application in MDTC (Chen & Cardie, 2018; Wu & Guo, 2020; Wu et al., 2022b). Traditional adversarial MDTC models typically comprise four components: (1) a shared feature extractor $F_s$, (2) a collection of domain-specific feature extractors $\{F_d^i\}_{i=1}^M$, (3) a classifier $C$, and

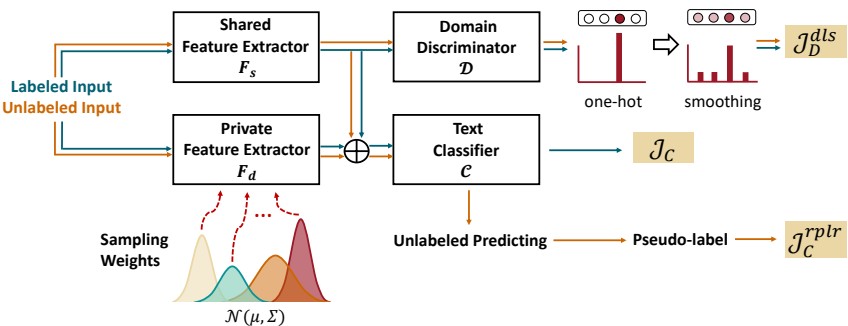

Figure 1: The architecture of the SAN method.

(4) a domain discriminator $D$. The objective of $F_s$ is to learn domain-invariant features capable of generalizing across diverse domains, while $\{F_d^i\}_{i=1}^M$ are designed to capture domain-unique features advantageous within their respective domains. $C$ serves as a binary classifier for sentiment prediction, while $D$ acts as an M-way classifier for domain identification. The feature extractors can adopt various neural network architectures, such as convolutional neural networks (CNNs) (Zhang et al., 2015), multi-layer perceptrons (MLPs) (Chen & Cardie, 2018), and transformers (Vaswani et al., 2017), to generate fixed-length feature representations. $D$ takes the shared feature vector as input, while $C$ takes the concatenation of the shared feature vector and the domain-specific feature vector. In conventional MDTC approaches, two primary objectives must be achieved: (1) minimizing the classification loss on labeled data, and (2) optimizing the adversarial loss on both labeled and unlabeled data. These objectives can be formulated as follows:

$$\min_{F_s, \{F_d^i\}_{i=1}^M, C} \max_D \mathcal{J}_C(F_s, \{F_d^i\}_{i=1}^M, C) + \lambda \mathcal{J}_D(F_s, D) \tag{1}$$

$$\mathcal{J}_C(F_s, \{F_d^i\}_{i=1}^M, C) = \sum_{i=1}^M \mathbb{E}_{(\mathbf{x}, y) \sim \mathbb{L}_i} [\mathcal{L}(C[F_s(\mathbf{x}), F_d^i(\mathbf{x})], y)] \tag{2}$$

$$\mathcal{J}_D(F_s, D) = \sum_{i=1}^M \mathbb{E}_{\mathbf{x} \sim \mathbb{L}_i \cup \mathbb{U}_i} [\mathcal{L}(D(F_s(\mathbf{x})), d)] \tag{3}$$

Where $\mathcal{L}(\cdot, \cdot)$ is the canonical classification loss, $[\cdot, \cdot]$ represents the concatenation of two vectors, and $d$ is the ground-truth domain label of the corresponding instance $\mathbf{x}$.

## 3.2 STOCHASTIC ADVERSARIAL NETWORK

Given that feature extractors typically employ intricate neural network architectures to capture valuable information from input data, and MDTC models necessitate training domain-specific feature extractors for each domain, this approach leads to a significant increase in model parameter count and a slowdown in convergence speed. To overcome this problem, we propose the stochastic adversarial network (SAN) for MDTC, which introduces a stochastic feature extractor to replace multiple domain-specific feature extractors without compromising model performance. The architecture of our proposed SAN method is depicted in Figure 1. The fundamental concept behind our approach is to model a distribution of domain-specific feature extractors, where the domain-specific feature extractors utilized to learn domain-unique features are simply random samples drawn from this distribution. This design permits access to an infinite number of domain-specific feature extractors, as we can sample any desired quantity of them. Furthermore, it decouples the number of domain-specific feature extractors from the model parameter count, ensuring that the model size remains unchanged as new domains emerge.

More specifically, we employ a multivariate Gaussian distribution $\mathcal{N}(\mu, \Sigma)$, where $\mu$ represents the mean vector and $\Sigma$ corresponds to the diagonal covariance matrix. The parameters of the domain-specific feature extractors for each domain can be randomly drawn from $\mathcal{N}(\mu, \Sigma)$. The resulting

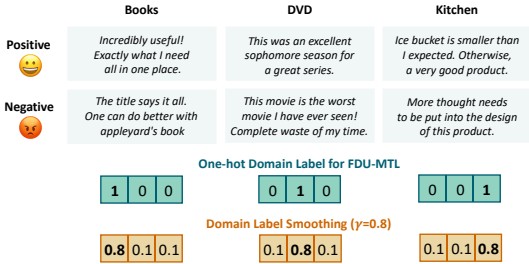

Figure 2: An example of DLS with three domains on the FDU-MTL dataset.

loss is then back-propagated to update the learnable parameters $\mu$ and $\Sigma$. It is important to note that inferring and training a neural network that models the domain-specific feature extractor as a weight distribution pose significant challenges. The random sampling process impedes conventional end-to-end training. To overcome this, we utilize the reparameterization technique (Kingma & Welling, 2013) to train the multivariate Gaussian representing the domain-specific feature extractor distribution. This enables the SAN model to be trained effectively using backpropagation. Specifically, we express the last fully connected layer of the domain-specific feature extractor as $\phi_d$, which is obtained as $\phi_d = \mu + \sigma \odot \epsilon$, where $\epsilon$ is an independent sample drawn from a standard Gaussian, $\odot$ denotes element-wise multiplication, and $\sigma$ represents the diagonal elements of $\Sigma$.

By adopting the stochastic domain-specific feature extractor, we can update Eq. 2 as:

$$\mathcal{J}_C(F_s, F_d, C) = \sum_{i=1}^{M} \mathbb{E}_{(\mathbf{x},y)\sim\mathbb{L}_i}[\mathcal{L}(C[F_s(\mathbf{x}), F_d(\mathbf{x})], y)] \tag{4}$$

Utilizing the stochastic feature extractor in our SAN method enables us to achieve competitive outcomes when compared to multinomial adversarial networks (Chen & Cardie, 2018) (As shown in Sec. D.2), while also significantly reducing the number of model parameters (As shown in Sec. D.5). In our study, we represent the stochastic domain-specific feature extractor as $F_d$. To enhance the performance of our model further, we integrate domain label smoothing (Zhang et al., 2023) and robust pseudo-label regularization (Gu et al., 2020). These additions serve to stabilize the adversarial training and enhance the discriminability of features, respectively.

### 3.3 Enhancement via domain label smoothing

Despite AT has been empirically proven effective in minimizing domain divergence and capturing domain-invariant features (Ganin et al., 2016; Chen & Cardie, 2018), it is widely acknowledged that AT is challenging to train and converge (Roth et al., 2017; Jenni & Favaro, 2019; Arjovsky & Bottou, 2017). This difficulty arises from the use of one-hot domain labels in AT, which leads to highly over-confident output probabilities. Consequently, the over-confidence of the domain discriminator can result in significant oscillatory gradients (Arjovsky & Bottou, 2017; Mescheder et al., 2018), negatively impacting training stability. To address this issue, we incorporate the domain label smoothing (DLS) technique, which encourages the domain discriminator to estimate soft probabilities instead of relying on confident classifications (Zhang et al., 2023). DLS achieves this by employing a weighted soft-encoding approach to represent domain labels (as depicted in Figure 2). The DLS formulation is as follows:

$$\mathcal{J}_D^{dls}(F_s, D) = \sum_{i=1}^{M} \mathbb{E}_{\mathbf{x}\sim\mathbb{L}_i\cup\mathbb{U}_i}[\gamma log(\mathcal{D}_i(F_s(\mathbf{x}))) + \frac{1-\gamma}{M-1} \sum_{j=1,j\neq i}^{M} log(\mathcal{D}_j(F_s(\mathbf{x})))] \tag{5}$$

Where $\mathcal{D}_i$ gives the $i$-th dimension of the domain discriminator's output vector and $\gamma$ ($\gamma \in (0, 1)$) is a hyperparameter. The DLS is theoretically and empirically demonstrated to be capable of improving robustness to noisy domain labels, converging faster, attaining more stable training, and better

generalization performance without extra parameters and optimization steps. With Eq. 4 and Eq. 5, the overall training objective can be updated as:

$$\min_{F_s, F_d, C} \max_{D} \mathcal{J}_C(F_s, F_d, C) + \lambda \mathcal{J}_D^{dls}(F_s, D) \tag{6}$$

### 3.4 ENHANCEMENT VIA ROBUST PSEUDO-LABEL REGULARIZATION

In MDTC, a considerable portion of each domain consists of unlabeled data, making it intuitive to leverage pseudo-labels, i.e., estimated labels of unlabeled data, to enhance feature discriminability. Nevertheless, since unlabeled data lack supervision, their pseudo-labels inevitably contain noise. To effectively select unlabeled data capable of generating reliable pseudo-labels and thereby improving feature discriminability, we integrate the robust pseudo-label regularization (RPLR) technique (Gu et al., 2020) into our proposed SAN method. The RPLR approach assesses the correctness of pseudo-labels for unlabeled data based on the feature distance to the corresponding class center in a spherical feature space. It treats incorrectly labeled data as outliers and models the conditional probability of outliers/inliers using a Gaussian-uniform mixture model. Specifically, $\hat{y}_j^u$ represents the generated pseudo-label for the input instance $\mathbf{x}_j^u$: $\hat{y}_j^u = \arg\max_{k}[C[F_s(\mathbf{x}_j^u), F_d(\mathbf{x}_j^u)]]_k$, where $[\cdot]_k$ denotes the $k$-th element. To model the fidelity of the generated pseudo-label, a random variable $z_j \in \{0, 1\}$ is introduced, indicating whether the data is correctly or incorrectly labeled with values of 1 and 0, respectively. Consequently, RPLR is formulated as follows:

$$\mathcal{J}_C^{rplr}(F_s, F_d, C, \phi) = \sum_{i=1}^{M} \mathbb{E}_{\mathbf{x}_j^u \sim \mathbb{U}_i}[w(\mathbf{x}_j^u)\mathcal{L}(C[F_s(\mathbf{x}_j^u), F_d(\mathbf{x}_j^u)], \hat{y}_j^u)] \tag{7}$$

$$w(\mathbf{x}_j^u) = \begin{cases} \beta_j & \text{if } \beta_j > 0.5 \\ 0 & \text{otherwise} \end{cases} \tag{8}$$

Where $\beta_j$ represents the probability of correctly labeled data, i.e., $\beta_j = Pr(z_j = 1|\mathbf{x}_j^u, \hat{y}_j^u)$. In this manner, unlabeled data with a probability of correct labeling below 0.5 are discarded. The posterior probability of correct labeling, i.e., $Pr(z_j = 1|\mathbf{x}_j^u, \hat{y}_j^u)$, is modeled by the feature distance between the data and the class center to which it belongs, using a Gaussian-uniform mixture model based on pseudo-labels. Given a feature vector $f_j^u = [F_s(\mathbf{x}_j^u), F_d(\mathbf{x}_j^u)]$ of an unlabeled instance $\mathbf{x}_j^u$, its distance to the corresponding class center $\mathcal{C}_{\hat{y}_j^u}$ for category $\hat{y}_j^u$ is calculated as:

$$d_j^u = \frac{f_j^u \cdot \mathcal{C}_{\hat{y}_j^u}}{\|f_j^u\| \|\mathcal{C}_{\hat{y}_j^u}\|} \tag{9}$$

The class center $\mathcal{C}_{\hat{y}_j^u}$ is defined in a spherical space as presented in (Gu et al., 2020), the details of computing $\mathcal{C}_{\hat{y}_j^u}$ are available in the Appendix. The distribution of feature distance $d_j^u$ is modeled by the Gaussian-uniform mixture model, a statistical distribution considering outliers (Coretto & Hennig, 2016; Lathuilière et al., 2018),

$$p(d_j^u|\hat{y}_j^u) = \pi_{\hat{y}_j^u}\mathcal{N}^+(d_j^u|0, \sigma_{\hat{y}_j^u}) + (1 - \pi_{\hat{y}_j^u})\mathcal{U}(0, \delta_{\hat{y}_j^u}) \tag{10}$$

Where $\mathcal{N}^+(d_j^u|0, \sigma)$ denotes a density function that is proportional to Gaussian distribution when $d_j^u \geq 0$, otherwise the density is zero. $\mathcal{U}(0, \delta_{\hat{y}_j^u})$ is uniform distribution defined on $[0, \delta_{\hat{y}_j^u}]$. Specifically, the Gaussian component captures the underlying probability distribution of correctly labeled data, while the uniform component provides a robust representation of the distribution for incorrectly labeled data. With equation 10, the posterior probability of correct labeling for unlabeled data $\mathbf{x}_j^u$ is defined:

$$\beta_j = \frac{\pi_{\hat{y}_j^u} \mathcal{N}^+(d_j^u|0, \sigma_{\hat{y}_j^u})}{p(d_j^u|\hat{y}_j^u)} \tag{11}$$

The parameters of Gaussian-uniform mixture models are $\phi = \{\pi_k, \sigma_k, \delta_k\}_{k=1}^K$ where $K$ is the number of classes. The details of approximating these parameters will be given in Sec. 3.5.

In summary, the ultimate optimization objective is defined as:

$$\min_{F_s, F_d, C} \max_D \mathcal{J}_C(F_s, F_d, C) + \lambda \mathcal{J}_D^{dls}(F_s, D) + \lambda_{rplr} \mathcal{J}_C^{rplr}(F_s, F_d, C, \phi) \tag{12}$$

## 3.5 Training procedure

In this section, we present how to optimize each component in the SAN model and estimate the parameters $\phi$ of Gaussian-uniform mixture models. To optimize the ultimate object in Eq. 12, we alternatively optimize the networks and estimate parameters $\phi$ by fixing other components following (Gu et al., 2020). We first initialize $F_s$, $F_d$, $C$, $D$ with Eq. 6 via training strategies as in (Chen & Cardie, 2018), then we take the following two steps to make the optimization.

**(1) Estimating $\phi$ with fixed $F_s$, $F_d$, $C$, $D$.** Fixing the parameters of $F_s$, $F_d$, $C$, $D$, we generate the pseudo-label $\hat{y}_j^u$ and calculate the distance $d_j^u$ for all unlabeled data, then $\phi$ is estimated using EM algorithm as below. Let $\tilde{d}_j^u = (-1)_j^m d_j^u$, where $m_j$ is sampled from Bernoulli distribution $B(1, 0.5)$, and $N^u$ denotes the number of unlabeled data, then $\phi$ can be estimated as follows:

$$\beta_j^{l+1} = \frac{\pi_{\hat{y}_j^u}^l \mathcal{N}(\tilde{d}_j^u|0, \sigma_{\hat{y}_l}^l)}{\pi_{\hat{y}_j^u}^l \mathcal{N}(\tilde{d}_j^u|0, \sigma_{\hat{y}_j^u}^l) + (1 - \pi_{\hat{y}_j^u}^l)\mathcal{U}(-\delta_{\hat{y}_j^u}^l, \delta_{\hat{y}_j^u}^l)}$$

$$\pi_k^{l+1} = \frac{1}{\sum_{j=1}^{N^u} I_{\{\hat{y}_j^u=k\}}} \sum_{j=1}^{N^u} I_{\{\hat{y}_j^u=k\}} \beta_j^{l+1}$$

$$\sigma_j^{l+1} = \frac{\sum_{j=1}^{N^u} I_{\{\hat{y}_j^u=k\}} \beta_j^{l+1} (\tilde{d}_j^u)^2}{\sum_{j=1}^{N^u} I_{\{\hat{y}_j^u=k\}} \beta_j^{l+1}}, \delta_k^{l+1} = \sqrt{3(q_2 - q_1^2)}$$

Where

$$q_1 = \frac{1}{\sum_{j=1}^{N^u} I_{\{\hat{y}_j^u=k\}} \beta_j^{l+1}} \sum_{j=1}^{N^u} \frac{1 - \beta_j^{l+1}}{1 - \pi_k^{l+1}} I_{\{\hat{y}_j^u=k\}} \tilde{d}_j^u$$

$$q_2 = \frac{1}{\sum_{j=1}^{N^u} I_{\{\hat{y}_j^u=k\}} \beta_j^{l+1}} \sum_{j=1}^{N^u} \frac{1 - \beta_j^{l+1}}{1 - \pi_k^{l+1}} I_{\{\hat{y}_j^u=k\}} (\tilde{d}_j^u)^2$$

We refer our readers to Gu et al. (2020) for the deduction details of the parameters $\phi$.

**(2) Optimizing $F_s$, $F_d$, $C$, $D$ with fixed $\phi$.** Given current pseudo-labels and estimated $\phi$, we follow the standard MDTC training protocol (Chen & Cardie, 2018) to train $F_s$, $F_d$, $C$, $D$ with Eq. 12.

## 4 Experiment

### 4.1 Setup

**Datasets.** We conducted experiments on two benchmark datasets for MDTC: the Amazon review dataset (Blitzer et al., 2007) and the FDU-MTL dataset (Liu et al., 2017). The Amazon review

dataset comprises four domains: books, DVDs, electronics, and kitchen. Each domain consists of 2000 labeled data instances, with 1000 positive and 1000 negative examples. The data has been pre-processed into a bag-of-features representation, which includes unigrams and bigrams, without preserving word order information. The FDU-MTL dataset reflects real-world scenarios and contains raw text data. It encompasses 14 product review domains, including books, electronics, DVDs, kitchen, apparel, camera, health, music, toys, video, baby, magazine, software, sport, as well as two movie review domains: IMDB and MR. Each domain includes a validation set of 200 samples and a test set of 400 samples. The training and unlabeled sets vary in size across domains, but generally consist of approximately 1400 and 2000 instances, respectively.

**Implementation details.** To ensure a fair comparison, we adopt identical network architectures as presented in MAN (Chen & Cardie, 2018). It is worth noting that we only replace the last fully connected layer of the domain-specific feature extractor with a stochastic layer. For the Amazon review dataset, we select the 5000 most frequent features and represent each review as a 5000-dimensional vector, where the feature values represent raw counts. Our feature extractors employ multi-layer perceptrons (MLPs) with an input size of 5000. Each feature extractor consists of two hidden layers with sizes of 1000 and 500, respectively. In the case of the FDU-MTL dataset, we employ a single-layer convolutional neural network (CNN) as the feature extractor. The CNN utilizes different kernel sizes (3, 4, 5) with a total of 200 kernels. The input to the CNN is a 100-dimensional embedding obtained by processing each word of the input sequence using word2vec (Mikolov et al., 2013). For all experiments, we set the batch size to 8, the dropout rate for each component to 0.4, and the learning rate of the Adam optimizer (Kingma & Ba, 2014) to 0.0001. The size of the shared features is set to 128, and the size of the domain-specific features is set to 64. Both the classifier and discriminator are MLPs with hidden layer sizes matching their respective inputs (128+64 for the classifier and 128 for the domain discriminator). Furthermore, we set the hyperparameters $\lambda$ to 0.0001, $\gamma$ to 0.9, and $\lambda_{rplr}$ to 1.

**Comparison methods.** In the MDTC tasks, we evaluate the SAN method against several state-of-the-art methods: The multi-task convolutional neural network (MT-CNN) (Collobert & Weston, 2008). The muti-task deep neural network (MT-DNN) (Liu et al., 2015). The collaborative multi-domain sentiment classification method (CMSC) trained with the least square loss (CMSC-LS), the hinge loss (CMSC-SVM), and the log loss (CMSC-Log) (Wu & Huang, 2015). The pre-trained BERT-base model fine-tuned on each domain (BERT) (Devlin et al., 2018). The adversarial multi-task learning for text classification method (ASP-MTL) (Liu et al., 2017). The multinomial adversarial network (MAN) trained with the least square loss (MAN-L2) and the negative log-likelihood loss (MAN-NLL) (Chen & Cardie, 2018). The dynamic attentive sentence encoding method (DA-MTL) (Zheng et al., 2018). The global and local shared representation-based dual-channel multi-task learning method (GLR-MTL) (Su et al., 2020). The conditional adversarial network (CAN) (Wu et al., 2021a). The co-regularized adversarial learning method (Wu et al., 2022a). For MS-UDA experiments, the baselines involve the marginalized denoising autoencoder (mSDA) (Chen et al., 2012), the domain adversarial neural network (Ganin et al., 2016), the multi-source domain adaptation network (MDAN) (Wu et al., 2021b), the MAN (MAN-L2 and MAN-NLL) (Chen & Cardie, 2018), the CAN (Wu et al., 2021a) and CRAL (Wu et al., 2022a).

Table 2: MDTC results on the Amazon review dataset

| Domain | CMSC-LS | CMSC-SVM | CMSC-Log | MAN-L2 | MAN-NLL | CAN | CRAL | SAN(ours) |
|--------|---------|----------|----------|--------|---------|-----|------|-----------|
| Books | 82.10 | 82.26 | 81.81 | 82.46 | 82.98 | 83.76 | 85.26 | **86.29 ± 0.26** |
| DVD | 82.40 | 83.48 | 83.73 | 83.98 | 84.03 | 84.68 | 85.83 | **86.43 ± 0.38** |
| Electr. | 86.12 | 86.76 | 86.67 | 87.22 | 87.06 | 88.34 | 89.32 | **89.78 ± 0.12** |
| Kit. | 87.56 | 88.20 | 88.23 | 88.53 | 88.57 | 90.03 | **91.60** | 91.31±0.15 |
| AVG | 84.55 | 85.18 | 85.11 | 85.55 | 85.66 | 86.70 | 88.00 | **88.45 ± 0.08** |

## 4.2 RESULT

**Multi-Domain Text Classification.** The experimental results on the Amazon review dataset and FDU-MTL dataset are reported in Table 2 and Table 3, respectively. We report the classification results of mean ± variance over five random runs. From Table 2, it can be noted that the SAN method obtains the best classification accuracy on 3 out of 4 domains, and yield state-of-the-art results for the average classification accuracy. For the experimental results on FDU-MTL, shown in Table 3, the

Table 3: MDTC results on the FDU-MTL dataset

| Domain | MT-CNN | MT-DNN | ASP-MTL | BERT | MAN-L2 | MAN-NLL | DA-MTL | GLR-MTL | SAN(ours) |
|---|---|---|---|---|---|---|---|---|---|
| books | 84.5 | 82.2 | 84.0 | 87.0 | 87.6 | 86.8 | 88.5 | 88.3 | **90.5 ± 0.3** |
| electronics | 83.2 | 88.3 | 86.8 | 88.3 | 87.4 | 88.8 | 89.0 | **90.3** | 87.7±0.6 |
| dvd | 84.0 | 84.2 | 85.5 | 85.6 | 88.1 | 88.6 | 88.0 | 87.3 | **89.7 ± 0.5** |
| kitchen | 83.2 | 80.7 | 86.2 | **91.0** | 89.8 | 89.9 | 89.0 | 89.8 | 90.4±0.9 |
| apparel | 83.7 | 85.0 | 87.0 | **90.0** | 87.6 | 87.6 | 88.8 | 88.2 | 87.4±0.7 |
| camera | 86.0 | 86.2 | 89.2 | 90.0 | **91.4** | 90.7 | 91.8 | 89.5 | 91.1±0.6 |
| health | 87.2 | 85.7 | 88.2 | 88.3 | 89.8 | 89.4 | 90.3 | **90.5** | 90.3±0.3 |
| music | 83.7 | 84.7 | 82.5 | 86.8 | 85.9 | 85.5 | 85.0 | **87.5** | 85.9±0.8 |
| toys | 89.2 | 87.7 | 88.0 | 91.3 | 90.0 | **90.4** | 89.5 | 89.8 | 90.3±0.7 |
| video | 81.5 | 85.0 | 84.5 | 88.0 | 89.5 | 89.6 | 89.5 | **90.8** | 90.0±0.5 |
| baby | 87.7 | 88.0 | 88.2 | 91.5 | 90.0 | 90.2 | 90.5 | **92.3** | 90.7±0.8 |
| magazine | 87.7 | 89.5 | 92.2 | **92.8** | 92.5 | 92.9 | 92.0 | 92.3 | 92.3±0.1 |
| software | 86.5 | 85.7 | 87.2 | 89.3 | 90.4 | 90.9 | 90.8 | **91.8** | 89.5±0.4 |
| sports | 84.0 | 83.2 | 85.7 | 90.8 | 89.0 | 89.0 | **89.8** | 87.8 | **90.0 ± 0.2** |
| IMDb | 86.2 | 83.2 | 85.5 | 85.8 | 86.6 | 87.0 | 89.8 | 87.5 | 89.3±0.7 |
| MR | 74.5 | 75.5 | **76.7** | 74.0 | 76.1 | 76.7 | 75.5 | 72.7 | 76.5±0.9 |
| AVG | 84.5 | 84.3 | 86.1 | 88.1 | 88.2 | 88.4 | 88.2 | 88.5 | **88.8 ± 0.1** |

Table 4: Multi-source unsupervised domain adaptation results on the Amazon review dataset

| Domain | mSDA | DANN | MDAN(H) | MDAN(S) | MAN-L2 | MAN-NLL | CAN | CRAL | SAN(Ours) |
|---|---|---|---|---|---|---|---|---|---|
| Books | 76.98 | 77.89 | 78.45 | 78.63 | 78.45 | 77.78 | 78.91 | **82.49** | 81.48 |
| DVD | 78.61 | 78.86 | 77.97 | 80.65 | 81.57 | 82.74 | 83.37 | 84.30 | **85.53** |
| Electr. | 81.98 | 84.91 | 84.83 | 85.34 | 83.37 | 83.75 | 84.76 | 86.82 | **87.12** |
| Kit. | 84.26 | 86.39 | 85.80 | 86.26 | 85.57 | 86.41 | 86.75 | **89.08** | 89.00 |
| AVG | 80.46 | 82.01 | 81.76 | 82.72 | 82.24 | 82.67 | 83.45 | 85.67 | **85.78** |

proposed SAN method outperforms MT-CNN and MT-DNN consistently across all domains with notable large performance gains. When compared with the state-of-the-art MAN-L2, MAN-NLL, DA-MTL, and GLR-MTL, SAN achieves competitive results in terms of average classification accuracy. The experimental results on both benchmarks validate the efficacy of our proposed method.

**Multi-Source Unsupervised Domain Adaptation.** In real application scenarios, it is not uncommon for the target domain to lack annotated data. Evaluating MDTC models under such circumstances is of utmost significance. In the multi-source unsupervised domain adaptation (MS-UDA) setting, we have multiple source domains, each containing both labeled and unlabeled data, and a target domain with only unlabeled data. Our MS-UDA experiments are conducted on the Amazon review dataset, following the same protocol as outlined in Chen & Cardie (2018). Specifically, in each experiment, three out of four domains were treated as source domains, while the remaining domain was treated as the target domain. As shown in Table 4, the proposed SAN method outperforms other baselines on two out of four domains as well as the average accuracy. It reveals that our SAN method has a good capacity for transferring knowledge to unseen domains. Further experimental results, including parameter sensitivity analysis, ablation study, convergence analysis, model runtime comparison and model parameter comparison, can be found in the Appendix.

## 5 CONCLUSION

In this paper, we propose stochastic adversarial networks (SANs) for multi-domain text classification. In contrast to previous MDTC models that rely on multiple domain-specific feature extractors to capture domain-unique features, we introduce a multivariate Gaussian distribution $\mathcal{N}(\mu, \Sigma)$ over the weights of the domain-specific feature extractor. This allows for the sampling of an arbitrary number of diverse domain-specific feature extractors, providing the ability to leverage an infinite number of such extractors without increasing the model size. Furthermore, we integrate domain label smoothing and robust pseudo-label regularization techniques to stabilize the adversarial training process and enhance feature discriminability. Experimental results on two MDTC benchmarks demonstrate the effectiveness of our SAN model in improving system performance on these benchmarks and its generalization ability to unseen domains.

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

## A DATASET

The experiments are conducted on two benchmarks: The Amazon review dataset (Blitzer et al., 2007) and the FDU-MTL dataset (Liu et al., 2017). The statistical details of these two datasets are presented in Table 5 and 6, respectively.

Table 5: Details of the Amazon review dataset

| Domain | Labeled | Unlabeled | Class. |
|---|---|---|---|
| Books | 2000 | 4465 | 2 |
| DVD | 2000 | 5681 | 2 |
| Electronics | 2000 | 3586 | 2 |
| Kitchen | 2000 | 5945 | 2 |

## B CENTER OF SAMPLES ON SPHERE

This section computes the class center of spherical samples used for robust pseudo-label regularization. Before computing the class centers on sphere, we begin by normalizing the concatenations of the shared features and domain-specific features. Let $f(\mathbf{x}) = [F_s(\mathbf{x}), F_d(\mathbf{x})]$, where $[\cdot, \cdot]$ represents the concatenation of two vectors. We then normalize features with $f' = r\frac{f(\mathbf{x})}{f(\mathbf{x})}$ to obtain features in the spherical space $\mathbb{S}_r^{n-1} = \{f' \in \mathbb{R}^n : ||f'|| = r\}$. Let $f'_1, f'_2, \cdots, f'_m$ be samples on the sphere $\mathbb{S}_r^{n-1}$,

Table 6: Details of the FDU-MTL dataset

| Domain | Train | Dev. | Test | Unlabeled | Avg. L | Vocab. | Class. |
|---|---|---|---|---|---|---|---|
| Books | 1400 | 200 | 400 | 2000 | 159 | 62K | 2 |
| Electronics | 1398 | 200 | 400 | 2000 | 101 | 30K | 2 |
| DVD | 1400 | 200 | 400 | 2000 | 173 | 69K | 2 |
| Kitchen | 1400 | 200 | 400 | 2000 | 89 | 28K | 2 |
| Apparel | 1400 | 200 | 400 | 2000 | 57 | 21K | 2 |
| Camera | 1397 | 200 | 400 | 2000 | 130 | 26K | 2 |
| Health | 1400 | 200 | 400 | 2000 | 81 | 26K | 2 |
| Music | 1400 | 200 | 400 | 2000 | 136 | 60K | 2 |
| Toys | 1400 | 200 | 400 | 2000 | 90 | 28K | 2 |
| Video | 1400 | 200 | 400 | 2000 | 156 | 57K | 2 |
| Baby | 1300 | 200 | 400 | 2000 | 104 | 26K | 2 |
| Magazine | 1370 | 200 | 400 | 2000 | 117 | 30K | 2 |
| Software | 1315 | 200 | 400 | 475 | 129 | 26K | 2 |
| Sports | 1400 | 200 | 400 | 2000 | 94 | 30K | 2 |
| IMDB | 1400 | 200 | 400 | 2000 | 269 | 44K | 2 |
| MR | 1400 | 200 | 400 | 2000 | 21 | 12K | 2 |

the center $\mathcal{C}$ of the samples on the sphere corresponds to the point closest to all samples, i.e., the solution of the following optimization problem:

$$\min_{f' \in \mathbb{S}_r^{n-1}} \frac{1}{m} \sum_{i=1}^{m} dist(f', f_i') \tag{13}$$

Where $dist(u,v) = 1 - \frac{u^T v}{||u|| ||v||}$ is the cosine distance. Since $||f'|| = r, \forall f' \in \mathbb{S}_r^{n-1}$, Eq. 13 can be rewritten as:

$$\max_{f'} f'^T (\sum_{i=1}^{m} f_i') \qquad s.t. ||f'|| = r. \tag{14}$$

With the method of Lagrange multipliers, the center can be obtained by:

$$\mathcal{C} = \frac{r}{||\tilde{f}'||} \tilde{f}' \tag{15}$$

Where $\tilde{f}' = \sum_{i=1}^{m} f_i'$.

## C  HOW STOCHASTIC FEATURE EXTRACTOR WORKS

The variance of the distribution $\Sigma$ elucidates the functioning of the stochastic feature extractor. As depicted in Figure 3, the initial values of $\Sigma$ follow a uniform distribution, whereas they exhibit more structured patterns after training. Notably, we can observe that the domain-specific feature extractor distribution tends to exhibit larger variances for distinct domains. As the SAN model converges, these significant variances ensure the distinct domain-unique features across domains.

## D  EXPERIMENTS

### D.1  PARAMETER SENSITIVITY ANALYSIS

In this section, we examine the sensitivity of our SAN method to the values of hyperparameters $\lambda$, $\gamma$ and $\lambda_{rplr}$. The $\lambda$ and $\lambda_{rplr}$ are evaluated in the range $\{0.0001, 0.05, 0.1, 0.5, 1, 10\}$ and

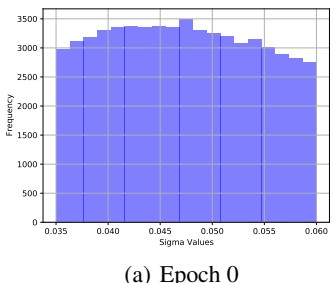

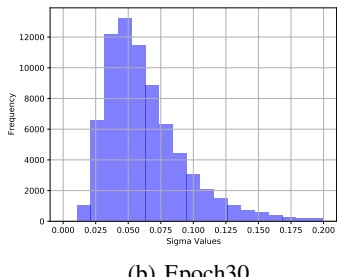

(a) Epoch 0              (b) Epoch30

Figure 3: The distribution of the flattened $\Sigma$ values for (a) Initialization (b) after convergence of SAN on the Amazon Review dataset.

$\{0.00001, 0.0001, 0.001, 0.01, 0.1, 1\}$, respectively. The valid range of values for $\gamma$ is $(0, 1]$, therefore we assess its impacts in the range of $\{0.5, 0.6, 0.7, 0.8, 0.9, 0.95\}$. We conduct the parameter sensitivity analysis on both the Amazon review dataset and FDU-MTL dataset. The experimental results are displayed in Figure 4 and Figure 5, respectively. We report the average classification accuracy.

In Figure 4, we observe a parameter sensitivity analysis performed on the Amazon review dataset. The results reveal that the impact of $\gamma$ on system performance can be negligible. Although the impact of $\lambda$ on system performance is relatively weak, increasing its value leads to a decline in classification accuracy. The most influential parameter is $\lambda_{rplr}$, with an increase in its value first leading to a rise in classification accuracy, followed by a decline. Figure 5 shows the results of the parameter sensitivity analysis conducted on the FDU-MTL dataset. In this case, we observe that no parameter has a significant effect on system performance, highlighting the stable training of our SAN method.

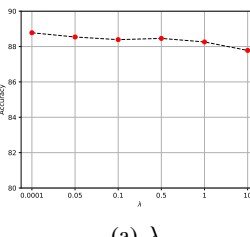
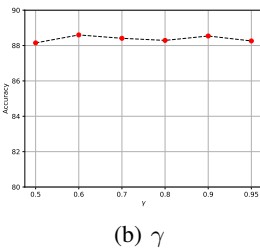
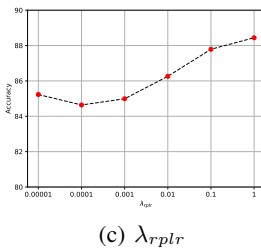

(a) $\lambda$            (b) $\gamma$            (c) $\lambda_{rplr}$

Figure 4: Parameter sensitivity analysis on Amazon review dataset.

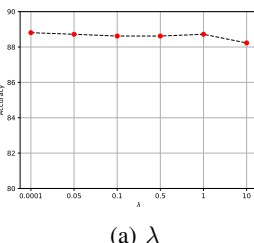
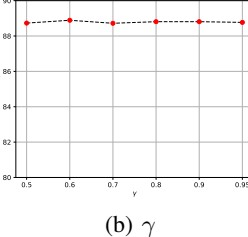
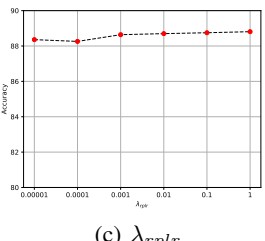

(a) $\lambda$            (b) $\gamma$            (c) $\lambda_{rplr}$

Figure 5: Parameter sensitivity analysis on FDU-MTL dataset.

## D.2 ABLATION STUDY

To assess the impact of each component of the SAN method on performance, we conduct an ablation study on the Amazon review dataset and FDU-MTL dataset. The experimental results are displayed in Figure 7 and Figure 8, respectively. Specifically, we examine three variants: (1) SAN w/o dls, a variant without the enhancement of domain label smoothing; (2) SAN w/o rplr, a variant without the enhancement of robust pseudo-label regularization; (3) plain SAN, a variant utilizing a stochastic feature extractor instead of the domain-specific feature extractors of MAN (Chen & Cardie, 2018). All three variants yield inferior results, while the full model achieves the best performance. The ablation study confirms the contributions of both components to the performance improvement of our model.

Table 7: Ablation study analysis on the Amazon review dataset

| Domain | SAN(full) | SAN w/o dls | SAN w/o rplr | plain SAN |
|--------|-----------|-------------|--------------|-----------|
| Books  | 86.29     | 84.70       | 83.05        | 82.25     |
| DVD    | 86.43     | 85.10       | 83.35        | 83.05     |
| Electr.| 89.78     | 89.75       | 87.75        | 86.90     |
| Kit.   | 91.31     | 90.85       | 88.15        | 88.25     |
| AVG    | 88.45     | 87.60       | 85.53        | 85.11     |

Table 8: Ablation Study on the FDU-MTL dataset

| Domain      | SAN(full) | SAN w/o dls | SAN w/o rplr | plain SAN |
|-------------|-----------|-------------|--------------|-----------|
| books       | 90.5      | 89.0        | 87.0         | 87.8      |
| electronics | 87.7      | 86.5        | 88.5         | 88.8      |
| dvd         | 89.7      | 90.0        | 90.8         | 88.3      |
| kitchen     | 90.4      | 90.3        | 90.5         | 89.8      |
| apparel     | 87.4      | 86.0        | 87.5         | 87.3      |
| camera      | 91.1      | 90.8        | 91.3         | 89.8      |
| health      | 90.3      | 90.5        | 90.0         | 91.3      |
| music       | 85.9      | 86.5        | 85.3         | 85.8      |
| toys        | 90.3      | 91.3        | 90.8         | 89.5      |
| video       | 90.0      | 90.3        | 88.3         | 89.5      |
| baby        | 90.7      | 90.8        | 90.0         | 90.0      |
| magazine    | 92.3      | 91.8        | 93.0         | 92.3      |
| software    | 89.5      | 89.0        | 90.5         | 89.0      |
| sports      | 90.0      | 88.0        | 88.8         | 90.3      |
| IMDb        | 89.3      | 89.8        | 88.8         | 86.5      |
| MR          | 76.5      | 76.3        | 74.3         | 72.0      |
| AVG         | 88.8      | 88.5        | 88.4         | 88.0      |

## D.3 CONVERGENCE ANALYSIS

We compare the convergence speed between our proposed SAN model and traditional MDTC methods employing the shared-private paradigm, such as MAN (Chen & Cardie, 2018). Figure 6 illustrates that our SAN approach exhibits a faster convergence rate compared to MAN, showcasing the accelerated speed achieved by the stochastic feature extractor.

## D.4 MODEL RUNTIME COMPARISON

We also compared the runtime of our proposed SAN model and traditional MDTC methods employing the shared-private paradigm (taking MAN as an example) on the Amazon review and FDU-MTL datasets, using the average training time per epoch as the indicator. The results, which are summarized in Table 9, are as follows: it is easy to observe that SAN requires less time, saving nearly 10% compared to MAN on the Amazon dataset and nearly 15% compared to MAN on the FDU-MTL dataset.

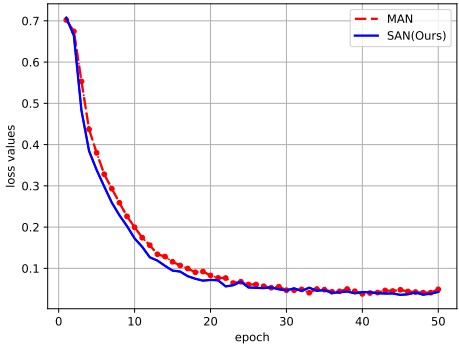

Figure 6: Convergence analysis between MAN and SAN.

Table 9: Model runtime comparison between MAN and SAN

| Task | Amazon | FDU-MTL |
|------|--------|---------|
| MAN | 7.07s | 70.88s |
| SAN(ours) | 6.39s | 60.72s |

## D.5 MODEL PARAMETER COMPARISON

We compare the conventional MDTC methods that employ the shared-private paradigm, such as MAN (Chen & Cardie, 2018), with our SAN approach in terms of the parameter counts of the shared feature extractor $F_s$, domain-specific feature extractors $\{F_d^i\}_{i=1}^M$, classifier $C$, and domain discriminator $D$. The results are presented in Table 10. The comparison reveals that in MAN, the domain-specific feature extractors contribute the most to the overall model parameters. In contrast, our proposed SAN method significantly reduces the parameter count of the domain-specific feature extractor while maintaining system performance. This further validates the effectiveness of our SAN approach. Moreover, the stochastic feature extractor we propose can seamlessly integrate into existing MDTC methods that adopt the shared-private paradigm.

Table 10: Model parameter comparison between MAN and SAN

| Dataset | Amazon | | FDU-MTL | |
|---------|--------|--------|---------|--------|
| Model | MAN | SAN(ours) | MAN | SAN(ours) |
| # Para. of $F_s$ | 5.57M | 5.57M | 20.20M | 20.20M |
| # Para. of $\{F_d^i\}_{i=1}^M$ | 22.13M | **5.57M** | 322.65M | **20.20M** |
| # Para. of $C$ | 0.04M | 0.04M | 0.04M | 0.4M |
| # Para. of $D$ | 0.02M | 0.02M | 0.02M | 0.02M |
| # Total Para. | 27.76M | **12.00M** | 342.91M | **40.46M** |

## E  LIMITATIONS

While our proposed SAN model demonstrates improved performance on the Amazon review dataset, its effectiveness on the FDU-MTL dataset falls short compared to state-of-the-art methods. In Table 11, we compare our SAN model with some most recent MDTC methods: the conditional adversarial network (CAN) (Wu et al., 2021a), the mixup regularized adversarial network (MRAN) (Wu et al., 2021b), the co-regularized adversarial network (CRAL) (Wu et al., 2022a), the robust contrastive alignment (RCA) (Li et al., 2022), and the maximum batch Frobenius norm (MBF) (Wu et al., 2022b). One of the main limitations of our work lies in the suboptimal accuracy of the pseudo-labels used for

robust pseudo-label regularization. In the SAN method, a pseudo-labeled data point $\mathbf{x}_j^u$ is considered valid when $w(\mathbf{x}_j^u)$ exceeds 0.5. The accuracy of valid pseudo-labels on unlabeled data in the Amazon review dataset is presented in Table 12. Additionally, Table 13 showcases the accuracy of valid pseudo-labels on the validation and test sets of the FDU-MTL dataset. Notably, the accuracy of pseudo-labels across different domains in the FDU-MTL dataset exhibits a significant variation, with the $'$MR$'$ domain achieving a mere 82.87% accuracy. It is worth mentioning that the integration of poor-quality pseudo-labels can substantially impair system performance. Therefore, we believe that enhancing the quality of pseudo-labels assigned to unlabeled data is crucial for improving our SAN approach.

Table 11: Comparisons of SAN with several state-of-the-art methods

| Method | CAN | MRAN | CRAL | MBF | RCA | SAN(ours) |
|---|---|---|---|---|---|---|
| Amazon | 87.70 | 87.64 | 88.00 | 87.71 | 86.88 | **88.45** |
| FDU-MTL | 89.4 | 89.0 | **90.2** | 90.1 | 89.0 | 88.8 |

Table 12: Accuracy of valid pseudo-labels on the Amazon review dataset

| Domain | Books | DVD | Elec. | Kit. | AVG |
|---|---|---|---|---|---|
| Acc. | 90.27 | 88.91 | 94.52 | 94.66 | $92.09 \pm 2.10$ |

Table 13: Accuracy of valid pseudo-labels on the FDU-MTL dataset

| Domain | Books | Elec. | DVD | Kit. | Apparel | Camera | Health | Music | Toys | Video | Baby | Magaz. | Softw. | Sports | IMDb | MR | AVG |
|---|---|---|---|---|---|---|---|---|---|---|---|---|---|---|---|---|---|
| Acc. | 89.67 | 94.63 | 90.69 | 95.21 | 96.05 | 94.84 | 93.98 | 87.76 | 93.12 | 90.25 | 93.99 | 95.82 | 92.17 | 95.88 | 89.33 | 82.87 | $92.27 \pm 3.52$ |

