# OpenReview forum: "Stochastic Adversarial Networks for Multi-Domain Text Classification"
_ICLR.cc/2024/Conference — ICLR 2024 Conference Withdrawn Submission_

### Official Review · Reviewer_agCZ · 2023-10-26

**Soundness:** 2 fair
**Presentation:** 2 fair
**Contribution:** 2 fair
**Rating:** 5
**Confidence:** 3

**Summary:**

To tackle Multi-Domain Text Classification (MDTC) task, one mainstream of proposed techniques is to extract the features via the shared and private extractors to capture the domain-invariant and domain-specific knowledge, respectively. However, as the number of domains increases, the count of their private extractors will also rapidly surge.
The author proposed a novel approach Stochastic Adversarial Network (SAN) to avoid the unaffordable explosion of parameters when encountering the newly emerged domains. Specifically, the author modeled the domain-specific feature extractors as a multivariate Gaussian distribution. Furthermore, some tricks, such as domain label smoothing and robust pseudo-label regularization techniques, are utilized to improve the overall performance.
Extensive experiments on two benchmarks demonstrate the superiority of the proposed method compared with the state-of-the-art baselines.

**Strengths:**

1.	This paper proposes a novel approach, called Stochastic Adversarial Network, to reduce the computational cost while meeting a large amount of domains.
2.	This paper originally employs Gaussian distribution to generate private extractors in order to circumvent the extensive parameters found in previous works.
3.	This paper conducts numerous experiments to show the effectiveness of the proposed scheme. Moreover, the parameter sensitivity and ablation study demonstrate the rationale of parameter selection and the necessity of each modules, respectively.

**Weaknesses:**

1.	The motivation is trivial. It is hard to say that the model size is the bottleneck of the training process according to Table.1 and 9. 342.91M is absolutely fine in current period. Further, inference process may gain nothing in the aspect of computational acceleration as we only choose one private extractor from the Domain Discriminator D.
2.	The baselines are outdated and improvements on two benchmarks are limited. According to Table 2,3 and 4, it can hardly convince me that the proposed model exactly outperforms the SOTA models. It is worth noting that the author points out this limitation in Appendix E.
3.	The writing and organization need to be improved.
a)	The emphasis in writing has been misplaced. As the author highlights the role of multivariate Gaussian distribution in Abstract, you are supposed to tell more story of it instead of the regularization term, which is the idea of others.
b)	The effectiveness is not the focus of this article, efficiency is. Therefore, moving D. 5 to the main body of the article perhaps make your contribution more prominent.
c)	Some tools can be utilized effectively to optimize sentence structure and composition.

**Questions:**

1.	The aim of equation (3) is to ensure that the shared Feature Extractor F_s exactly extract the domain-invariant features. Thus the author maximum this loss to let the discriminator D be confused about the features coming from F_s. Here is the question: discriminator D may lack of capabilities to recognize the difference among domains as this loss function does not involve any domain knowledge.
There may exists another adversarial network in equation (3), i.e. domain-specific extractor enhances the capabilities of discriminator D and domain-invariant extractor still confuse the discriminator D.
2.	As a classic NLP task, this method inevitably needs to be compared with chatgpt. Currently, chatgpt has shown remarkable zero-shot capabilities. Therefore, you need to convince the reviewers why your method should be used instead of chatgpt or highlight the scenarios in which your method has significant advantages.

---

### Official Review · Reviewer_NpVu · 2023-10-30

**Soundness:** 1 poor
**Presentation:** 3 good
**Contribution:** 1 poor
**Rating:** 1
**Confidence:** 4

**Summary:**

The paper presents a new model for MDTC, built on the previous shared-private feature extraction architecture. The innovation includes 1) modelling the parameter of domain-specific feature extractors as a Gaussian random variable, and for each domain, the parameter is drawn from the distribution. This is why the model is called stochastic adversarial network, or SAN, 2)  domain label smoothing 3) pseudo labelling regularization.  The authors show some empirical successes on some datasets.

**Strengths:**

The paper demonstrates that the authors are well aware of the challenges in MDTC and are familiar with various tools in deep learning (such as reparametrization trick, label smoothing, pseudo labelling etc).

**Weaknesses:**

I have some concerns about this work.

1. Assuming the design of proposed model is sensible (in fact I have doubts on this; see 2), the work heuristically puts together a bunch of well-known techniques to improve performance. Works of primarily such a nature, although potentially valuable in practice, do not possess enough novelty that justifies a publication in ICLR.

2. I have doubts on the proposed approach in the "stochastic" part. Let us track the parameter $W_1$ of the domain-specific feature extractor for domain 1. In the beginning it is drawn from the prescribed Gaussian, say, its value is $W_1^{(0)}$, and after the first iteration, the Gaussian parameter gets updated (using the reparametrization trick)  -- well, whether Gaussian parameter is updated or not is not critical here. Then in the next iteration, $W_1$  is drawn again, let us call it $W_1^{(1)}$. If this understanding is correct, then $W_1^{(0)}$ and $W_1^{(1)}$ can be very different. That is, along the training process, $W_1$ will randomly hop everywhere as long as the Gaussian variance is not vanishing. How would such a scheme work at all? Bringing the parameter $W_2$ of the second domain-specific extractor into the picture would show an even more absurd picture: at each iteration $t$, $W_1^{(t)}$ and  $W_2^{(t)}$ are random variables following the same Gaussian distribution. How would $W_1$ and $W_2$ track their respective domain specific features?  If this structure were to work, it would have to be the case where the Gaussian variance is very small (which might be the case as shown in Figure 3 of the appendix). In that case, all domain-specific extractors are more or less the same, i.e, all equal to the Gaussian mean, only subject to some tiny *domain-nonspecific* random perturbation. That would defeat the entire purpose of having domain specific feature extractors. -- I could misunderstood the paper and I am willing to hear the authors' defence on this. In your defence, please also show the initial and final values of the Gaussian mean vector $\mu$ (say, in terms of its L1-norm divided by its dimension), I would like compare it with $\sigma$.

**Questions:**

See weakness 2.

Additional question: The authors say that the conventional shared-private adversarial scheme will have "exponential increase" in model parameters as new domains emerge? Why is it exponential?

---

### Official Review · Reviewer_bAwA · 2023-11-01

**Soundness:** 3 good
**Presentation:** 3 good
**Contribution:** 2 fair
**Rating:** 5
**Confidence:** 2

**Summary:**

The paper tackles the multi-domain text classification (MDTC) problem, and tries to minimize the amount the learning parameters by introducing a stochastic feature extractor (domain feature). The model is effective in handling the benchmark datasets and outperform the other baseline models. Additional multi-source UDA experiment is also conducted as a simple model extension.

**Strengths:**

The proposed model performs strong in the benchmark dataset, with minimized learning parameters. The design of using both shared/private feature extractor is interesting and effective in merging the domain in the latent space. The proposed method is straightforward and easy to understand.

**Weaknesses:**

1. Though the proposal seems to be effective and achieving strong performance, the model itself still uses a relative old adversarial backbone, with the discriminator approach for removing the domain invariant feature. The two-feature-extractor approach is interesting, but that is mainly to deal with parameter increase in the MDTC problem. It would be great to see other design improvement in the model.
2. The performance gain in using the proposed model is marginal on the Amazon review/FDU-MTL datasets. Also, it would be great to have some analysis on adjusting the setting between the two feature extractors.

**Questions:**

1. This might be somewhat irrelevant, but would the model perform well in multi domain classification in other domain type(s), e.g., images?